# Diversity and asynchrony in soil microbial communities stabilizes ecosystem functioning

Cameron Wagg[1,2,3†]*, Yann Hautier[4†], Sarah Pellkofer[1,2†], Samiran Banerjee[2,5], Bernhard Schmid[1,6], Marcel GA van der Heijden[1,2,7]

[1]Department of Evolutionary Biology and Environmental Studies, University of Zürich, Zürich, Switzerland; [2]Plant-Soil Interactions, Research Division Agroecology and Environment, Agroscope, Zürich, Switzerland; [3]Fredericton Research and Development Centre, Agriculture and Agri-Food Canada, Fredericton, Canada; [4]Ecology and Biodiversity Group, Department of Biology, Utrecht University, Padualaan, Netherlands; [5]Department of Microbiological Sciences, North Dakota State University, Fargo, United States; [6]Department of Geography, Remote Sensing Laboratories, University of Zürich, Zürich, Switzerland; [7]Department of Plant and Microbial Biology, University of Zürich, Zürich, Switzerland

**Abstract** Theoretical and empirical advances have revealed the importance of biodiversity for stabilizing ecosystem functions through time. Despite the global degradation of soils, whether the loss of soil microbial diversity can destabilize ecosystem functioning is poorly understood. Here, we experimentally quantified the contribution of soil fungal and bacterial communities to the temporal stability of four key ecosystem functions related to biogeochemical cycling. Microbial diversity enhanced the temporal stability of all ecosystem functions and this pattern was particularly strong in plant-soil mesocosms with reduced microbial richness where over 50% of microbial taxa were lost. The stabilizing effect of soil biodiversity was linked to asynchrony among microbial taxa whereby different soil fungi and bacteria promoted different ecosystem functions at different times. Our results emphasize the need to conserve soil biodiversity for the provisioning of multiple ecosystem functions that soils provide to the society.

*For correspondence:
cameron.wagg@canada.ca

†These authors contributed equally to this work

## Introduction

The loss of biodiversity can compromise the functioning of ecosystems and services that the society depends upon (*Tilman and Downing, 1994*; *Hector and Bagchi, 2007*; *Cardinale, 2012*; *Hooper et al., 2012*). For example, evidence is mounting that the reduction in species diversity can reduce multiple ecosystem functions, such as processes that drive nutrient and carbon cycling and plant productivity (*Zavaleta et al., 2010*; *Maestre et al., 2012*; *Hautier et al., 2018*). Reduced biodiversity has also been shown to destabilize ecosystem functioning over time, and thus the maintenance of biodiversity is critical to long-term ecosystem sustainability (*Tilman et al., 2006*; *Hector et al., 2010*; *Isbell, 2011*; *Hautier, 2015*). Experiments with plant communities have shown that species diversity can support multiple ecosystem functions simultaneously because plant species that do not contribute to that function at a particular time may contribute at another time, or to another function at the same time. Thus, the asynchronous temporal fluctuations among species in their abundance and contribution to various ecosystem functions in more diverse communities contribute to a greater overall ecosystem functioning over time (*Hautier et al., 2018*; *Isbell, 2011*; *Yachi and Loreau, 1999*).

While the reduction of plant diversity is well known to destabilize ecosystem functioning, effects of soil biodiversity loss on stabilizing ecosystem functioning are still poorly understood. A number of studies have shown that soil biodiversity is rapidly declining in intensively managed soils (*Helgason et al., 1998*; *Birkhofer, 2008*; *Verbruggen, 2010*; *Tsiafouli et al., 2015*), and one-quarter of soils worldwide are now degraded, causing a reduction of biological productivity (*Stavi and Lal, 2015*). Thus, understanding the consequences of soil biodiversity loss is crucial, as there is a growing consensus that soil biota play key roles in ecosystems (*Bardgett and van der Putten, 2014*; *van der Heijden et al., 2008*; *Delgado-Baquerizo et al., 2016*) and support a number of ecosystem services, including food production and nutrient cycling (*Brussaard et al., 2007*; *Bender et al., 2016*). Several studies have demonstrated that the composition and richness of soil microbiota can predict multiple ecosystem functions, such as plant diversity and productivity, soil carbon assimilation, and nutrient cycling (*Bradford, 2014*; *Wagg et al., 2014*; *Jing et al., 2015*; *Delgado-Baquerizo et al., 2020*; *Mori et al., 2016*; *Wagg et al., 2019*). However, few studies have linked soil microbiota communities to the temporal stability of plant community composition and productivity (*Eisenhauer et al., 2012*; *Yang et al., 2014*; *Pellkofer et al., 2016*).

The compositions of soil bacterial and fungal communities are highly dynamic over short time scales, such as months and growing seasons, and temporal changes in composition are further altered by land management practices such as tillage and crop rotation (*Lauber et al., 2013*; *Coudrain et al., 2016*; *Wagg et al., 2018*). Whether these temporal dynamics in microbial communities lead to temporal fluctuations in ecosystem functions and if microbial community dynamics are inhibited due to soil microbial biodiversity loss have not been assessed yet (*Bardgett and van der Putten, 2014*). Past studies have shown that some soil functions do not track soil microbial loss, or they recover rapidly after perturbations of soil microbial communities. This suggests that such functions may either be highly resilient or contain many functionally redundant taxa (*Griffiths et al., 2000*; *Fitter et al., 2005*; *Allison and Martiny, 2008*). However, functional redundancy is likely to fade as multiple time points and ecosystem functions are considered (*Hector and Bagchi, 2007*; *Isbell, 2011*; *Hautier et al., 2018*). Considering that soil biodiversity supports numerous ecosystem functions, we hypothesize that greater soil biodiversity would also maintain a greater and less variable ecosystem functioning through time and would thus stabilize multiple ecosystem functions. Specifically, microbial taxa may fluctuate asynchronously in abundance through time such that different taxa support different functions at different times. This should provide insurance that some microbial taxa that support an ecosystem function will be present at any given time to stabilize the functioning of the ecosystem (*Isbell, 2011*; *Yachi and Loreau, 1999*; *Mori et al., 2016*). Therefore, we predict that temporal asynchrony among taxa in diverse soil microbial communities should relate to greater temporal stability in multiple ecosystem functions (*Loreau and de Mazancourt, 2008*; *Thibaut and Connolly, 2013*; *Gross et al., 2014*).

Here, we decreased soil diversity by filtering out soil organisms based on size along a gradient to assess the effects of soil microbial communities on stabilizing four ecosystem functions. The exclusion of organisms based on size can lead to a functional simplification of soil communities because body size is directly associated with trophic guilds, metabolic rates, population density, and generational turnover (*Coudrain et al., 2016*; *Bradford, 2002*; *Yodzis and Innes, 1992*; *Wall and Moore, 1999*; *Woodward, 2005*). Furthermore, the size-based reduction of soil organisms parallels the impact of land management practices, such as soil tillage, that physically damage soil organisms depending on their size and, thus, also disrupts the community structure of soil biota (*Wagg et al., 2018*; *Jansa et al., 2003*; *Köhl et al., 2014*; *Postma-Blaauw et al., 2010*).

## Results

Using soil sieving gradient (5000, 100, 25, and 0 µm [sterile]), we created a monotonically declining gradient in fungal and bacterial richness among replicated soil-plant ecosystems established in self-contained mesocosms ($F_{1, 46}$ = 172.0, p<0.001; and $F_{1, 45.7}$ = 753.9, p<0.001, respectively, for fungal and bacterial richness decline; *Figure 1—figure supplement 1*, *Supplementary file 1 – table 1*). The decline in diversity over our experimental gradient was maintained throughout the experimental duration (time [factor] by sieve gradient (log-linear) interaction: $F_{4, 170.1}$ = 0.75, p=0.561; and $F_{4, 177.1}$ = 1.60, p=0.174, respectively, for fungi and bacteria; *Figure 1—figure supplement 1*, *Supplementary file 1- table 1*). This decline in bacterial richness along the four-step gradient was

steepest for the final step from smallest sieve size to the sterile soil treatment), which resulted in a 60% loss of bacterial and 55% loss of fungal richness, compared to the highest soil biodiversity treatment. Community composition differed among the soil diversity treatments and the different time points, for both fungal and bacterial communities (*Figure 1—figure supplement 1* and *2*, *Supplementary file 1* – table 2). All four ecosystem functions (plant biomass production, plant diversity, litter decomposition, and soil carbon assimilation) changed significantly over our soil biodiversity treatment gradient and varied significantly through time depending on the soil biodiversity level (*Figure 1a–d*, *Supplementary file 1* – tables 3 and 4). Bacterial and fungal richness and community composition were significantly related to all four ecosystem functions, depending on the

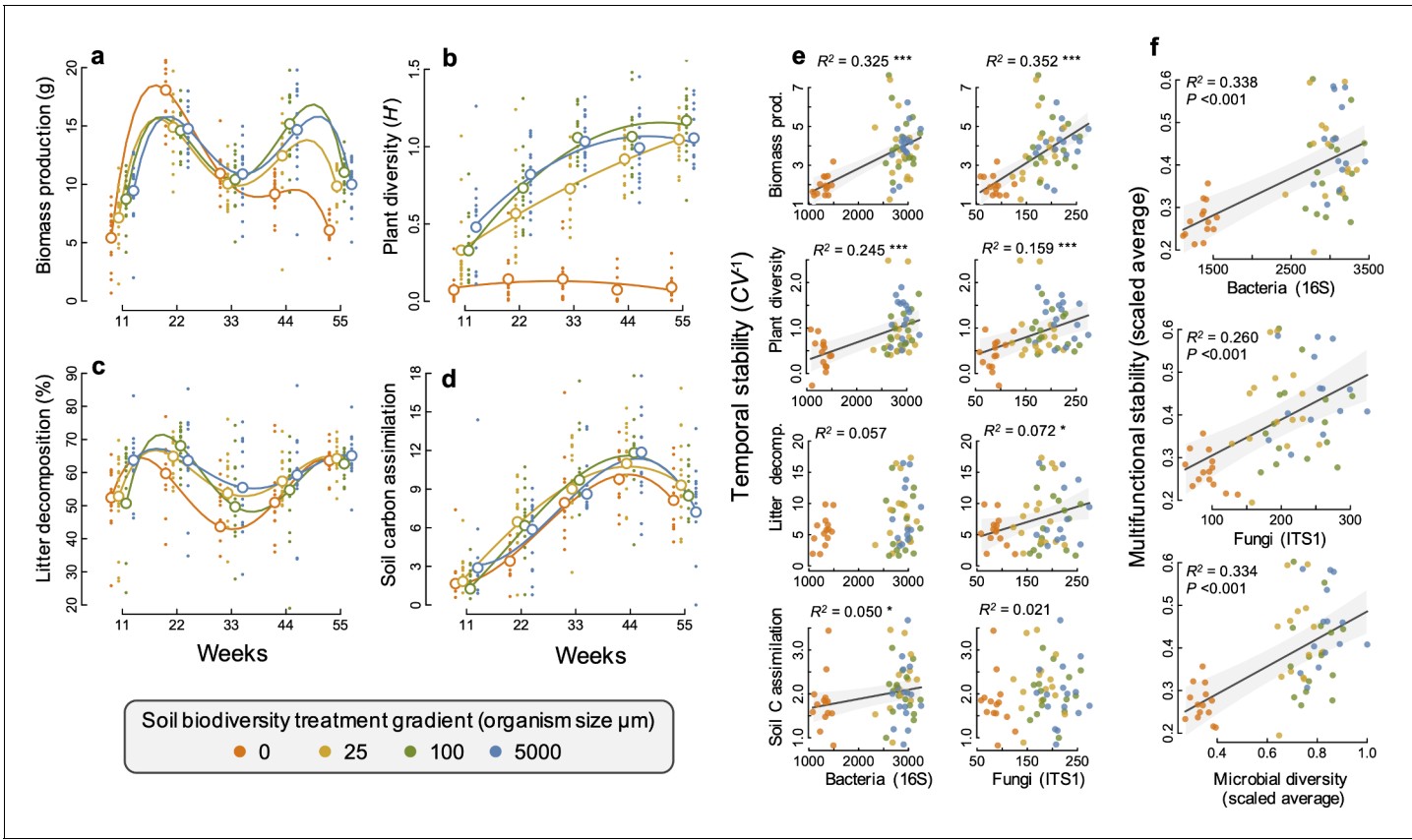

**Figure 1.** Changes in ecosystem functions through time and relationships between soil microbial diversity and multifunctional stability. (**a**) Plant biomass production, (**b**) plant diversity, (**c**) litter decomposition, and (**d**) soil carbon assimilation, measured as the $\delta^{13}C$ in the soil and scaled by the minimum value such that larger positive values indicate a greater amount of $\delta^{13}C$ detected in the soil. Individual points are data from individual mesocosms and larger open points are the means for each time point and soil biodiversity treatment level. Polynomial regression lines are shown to highlight the temporal trends. (**e**) The relationship between the temporal stability (the inverse coefficient of variation, $CV^{-1}$) of each of the four functions and the richness of bacteria and fungi. (**f**) Multifunctional stability, calculated as the scaled average of the stability of each of the four ecosystem functions shown in relation to the richness of bacteria, fungi, and microbial diversity (the scaled average of fungal and bacterial richness) are all highly significant. Regression lines and 95% confidence bands are shown for significant relationships (p<0.05).

The online version of this article includes the following figure supplement(s) for figure 1:

**Figure supplement 1.** Changes in soil microbial diversity for each treatment along the diversity gradient.

**Figure supplement 2.** Proportional abundance of (A) fungal classes and (B) bacterial classes is shown for each time point (11, 22, 33, 44, and 55 weeks) and for the different inoculum treatments created by sieving soil through 5000, 100, 25 μm, and sterile, corresponding to 0 μm (shown in subpanels from left to right).

**Figure supplement 3.** Bacterial richness-ecosystem function relationships for each time point.

**Figure supplement 4.** Fungal richness-ecosystem function relationships for each time point.

**Figure supplement 5.** Mantel correlation tests associating the dissimilarity in ecosystem functioning and the dissimilarity in (A) the fungal and (B) the bacterial community compositions at each of the five sampling times.

**Figure supplement 6.** Plant species proportional abundance.

time, and differences in microbial community composition were all correlated with differences in ecosystem functions at all time points (*Figure 1—figure supplement 3–5*). These results demonstrate that we successfully established a broad gradient of soil microbial diversity, composition, and ecosystem functioning. This enabled us to assess effects of the soil microbial community on the temporal stability of ecosystem functioning, which we calculated as the inverse of the coefficient of variation of ecosystem functions across all five time points (*Tilman et al., 2006*; *Pimm, 1984*; *Tilman, 1999*).

We found that greater bacterial and fungal richness supported higher stability of two aboveground ecosystem functions, plant productivity and plant diversity (*Figure 1e*). Plant biomass production was particularly strongly destabilized in our sterile treatment with the lowest soil microbial diversity, exhibiting a 'boom and bust' temporal response, where plant biomass was the lowest of all treatments after 11 weeks, then the highest at 22 weeks and steadily declined afterwards to substantially lower values than in all other treatments by 55 weeks (*Figure 1a*). The effects of microbial diversity on the stability of plant biomass production were also significant when the sterile treatment was not considered, showing that microbial diversity is also important for plant community stability even at higher levels of microbial diversity. All mesocosms were initially planted with the same plant community and plant diversity increased steadily over time as communities developed. This occurred in all but the sterile soil treatment, which remained low throughout the experiment. This resulted in the positive effect of microbial diversity on stabilizing plant diversity (*Figure 1b*).

The effect of soil microbial diversity on stabilizing plant biomass and diversity resulted from its influence on the changes in plant community composition. In the sterile and low soil biodiversity treatments, plant communities were dominated by the grass *Lolium perenne* (*Figure 1—figure supplement 6*) which drove the 'boom and bust' trend in biomass production (*Figure 1a*). In contrast, the biomass production by the legume *Trifolium pratense* increased steadily over time and was highest in the highest soil biodiversity treatment, contributing to the greater stability in plant diversity and productivity. The sterile treatment could account for the effect of microbial diversity on supporting more stable plant diversity, as well as the bacterial richness-biomass stability relationship, but not the stabilizing effect that fungal richness had on plant biomass production (*Supplementary file 1* – table 5).

Fungal richness was significantly related to greater stability in litter decomposition (*Figure 1e*), and soil carbon assimilation was positively related to greater bacterial richness (*Figure 1e*). Using an averaging multifunctionality approach, where multifunctionality is calculated as the standardized average of all ecosystem functions (*Maestre et al., 2012*; *Byrnes, 2014*), we found that greater microbial diversity (richness of bacteria, fungi, and their scaled average) all showed strong positive relationships with the ability to maintain greater and more stable ecosystem functioning on average (*Figure 1f*). Using a multiple threshold approach (*Byrnes, 2014*) to characterize how many functions achieve an increasing threshold of stabilization with greater diversity, we found that soil microbial diversity had a significant stabilizing effect on multiple functions at thresholds above 30% of the maximum stability value for each function (*Figure 2a*). At this threshold, more than three functions had high level of temporal stability (*Figure 2b*), showing that high soil biodiversity can promote the stability of multiple functions simultaneously. This stabilizing effect increased to become strongest at a threshold of 72%. Soil bacterial richness had a significant stabilizing effect on multiple functions at low threshold values of 29% and was strongest at 68% (*Figure 2c*). Similarly, soil fungal richness had a positive effect on the stability of multiple functions at 30% and had the strongest effect on supporting stability in multiple functions at 82% of the maximum observed stability (*Figure 2d*).

Since greater plant diversity is often associated with greater plant biomass production (*Tilman and Downing, 1994*; *Hector and Bagchi, 2007*; *Cardinale, 2012*; *Hooper et al., 2012*), and biomass production with soil carbon assimilation (*Lange et al., 2015*; *Yang et al., 2019*), we assessed how soil microbial diversity may have indirectly influenced the temporal stability of plant biomass production through its strong relationship with plant diversity using multi-model comparisons and structural equation modeling (SEM). We found that soil microbial diversity alone was the most parsimonious explanatory variable of plant biomass stability compared with models that included plant diversity (*Supplementary file 1* – table 6). Further, using SEM we found that soil microbial diversity reduced the temporal variation and increased the temporal mean in biomass production, while plant diversity had no detectable effect (*Figure 3a*). We also found that the temporal variation had a much stronger influence compared with the temporal mean on the stability of

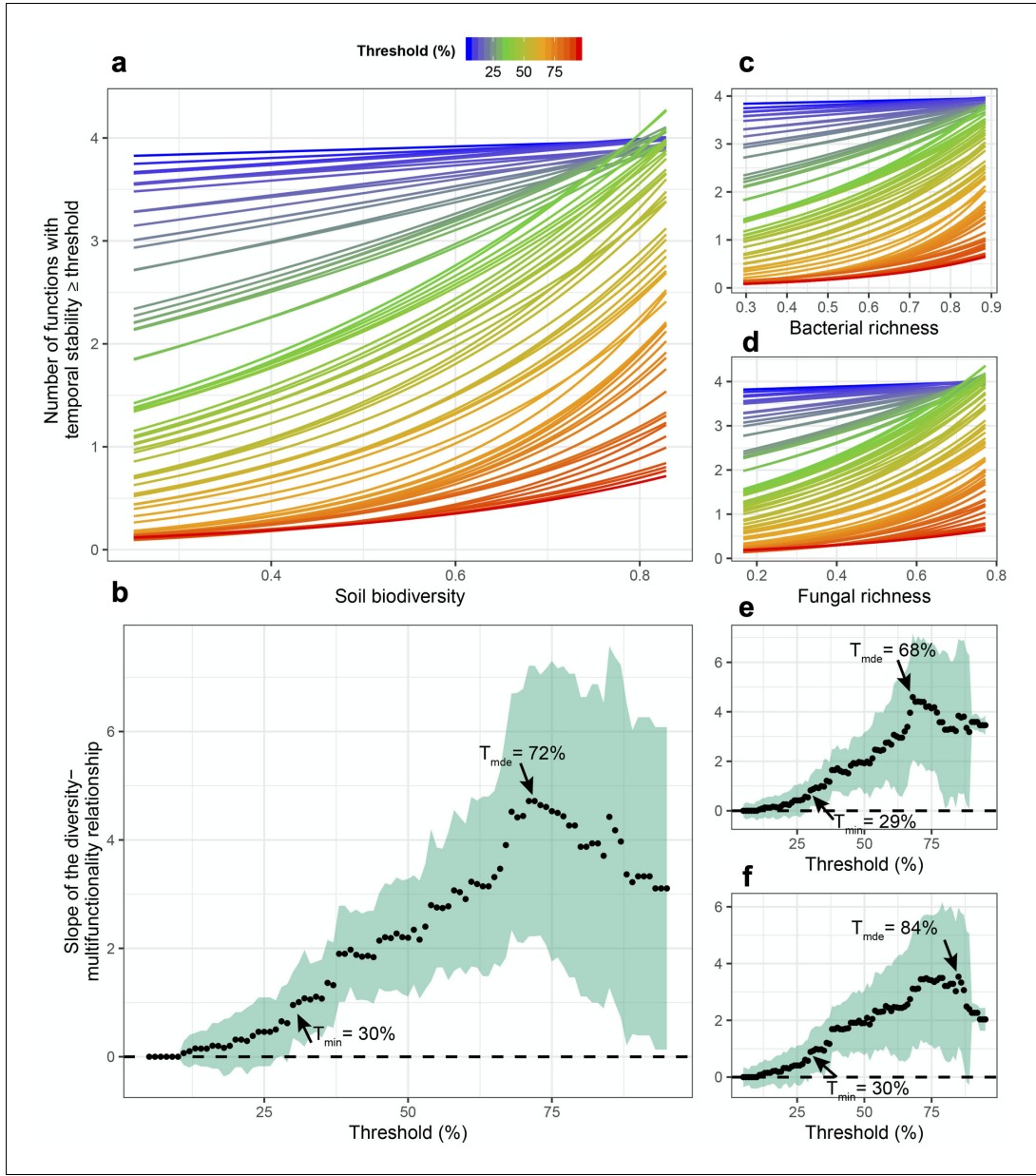

**Figure 2.** Soil microbial diversity effects on the temporal stability of ecosystem multifunctionality over a range of multifunctionality thresholds. (**a**) Slopes of soil microbial diversity (average of scaled bacterial and fungal richness) relationship with ecosystem multifunctional stability are shown over a range of thresholds. Points are the slopes of the relationships and the shaded green area represents the 95% confidence intervals around the slope such that diversity effect on the stability of multifunctionality is significant when the intervals do not overlap the zero dashed line. (**b**) The effects of soil microbial diversity on the number ecosystem functions they stabilze above a threshold. Bold lines represent the minimum threshold above which ($T_{min}$) and maximum threshold below which ($T_{max}$) stability of multifunctionality is associated with diversity, and the realized maximum diversity effect ($R_{mde}$) where the slope of the diversity-stability of multifunctionality relationship is steepest. (**c**) The slopes of the richness-multifunctional stability relationships and the number of ecosystem functions supported for each threshold are shown for bacterial richness. (**d**) The slopes of the richness-multifunctional stability relationships and the number of ecosystem functions supported for each threshold are shown for fungal richness.

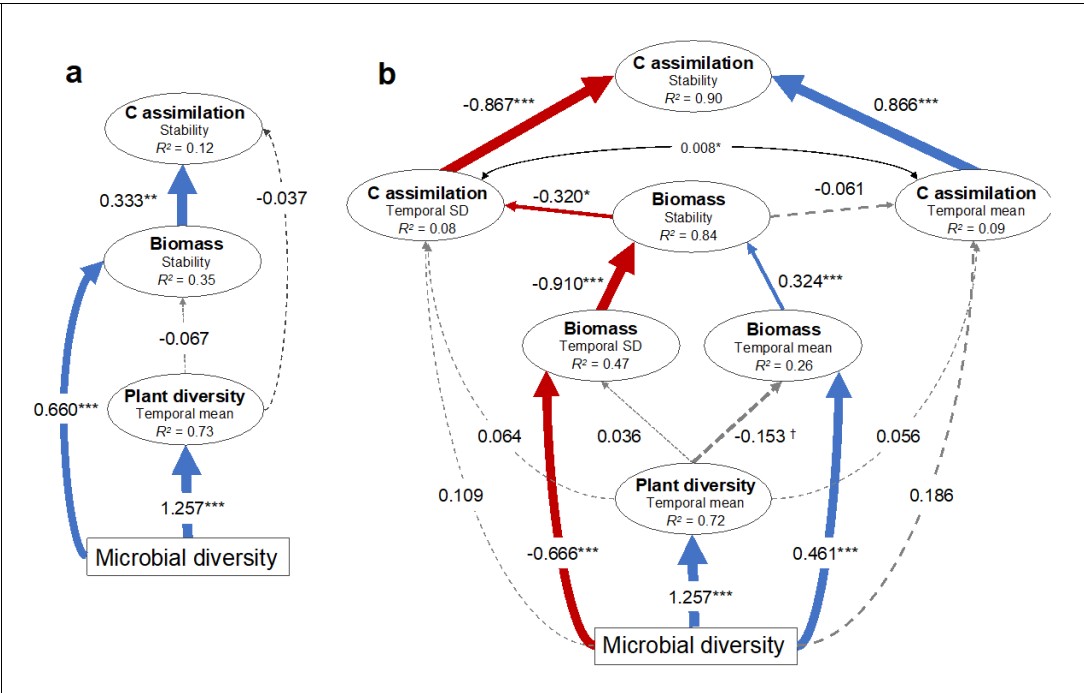

**Figure 3.** Structural equation models illustrating the indirect effects of soil microbial diversity on the stability of plant biomass production and soil carbon assimilation. (a) Model illustrating that microbial diversity provides greater stability in biomass production that in turn stabilizes soil carbon assimilation. The microbial diversity to carbon assimilation was the least significant path and was removed so that the model was not fully saturated. Model fit: Chi-square = 0.003, p=0.954; RSEMA (Root Mean Squared Error of approximation) = 0, p=0.957; SRMR (Standardized Root Mean Square Residual) = 0.001, BIC (Bayesian Information Criterion) = −122.2. (b) Since stability is the ratio of the temporal mean and variance (SD), the effects of microbial diversity on stability through affecting the temporal mean and SD are shown. The model indicates that microbial diversity promotes biomass stability both by increasing biomass and reducing the variation in biomass production. The mean and SD of soil carbon assimilation were allowed to covary (indicated by the double arrow). Model fit: Chi-square = 15.9, p=0.144; RMSEA = 0.086, p=0.241; SRMR (Standardized Root Mean Square Residual) = 0.033, BIC (Bayesian Information Criterion) = −568.7. See *figure supplement 1* for the effects of the temporal mean and variation of plant biomass production on the temporal mean and variation in soil carbon assimilation respectively that produced similar results. Red and blue arrows indicate negative and positive effects respectively and the standardized coefficients for each are show adjacent. Non-significant effects are indicated by grey dashed lines. Significance is indicated by †p<0.1, *p<0.05, **p<0.01, ***p<0.001. The marginal $R^2$ for each endogenous variable is provided (within ellipses).

The online version of this article includes the following figure supplement(s) for figure 3:

**Figure supplement 1.** Structural equation model (SEM) results illustrating the direct and indirect effects of microbial diversity on the stability in plant biomass production and stability in soil carbon assimilation.

biomass production (*Figure 3b*). Comparing multiple models using soil microbial diversity, plant diversity, plant biomass, and the stability in plant biomass to predict the stability in soil carbon assimilation, we found that stability in soil carbon assimilation was best explained by the stability in plant biomass production alone (*Supplementary file 1* – table 7). Therefore, the effect of soil microbial diversity on stabilized soil carbon assimilation was likely indirect through its stabilizing effect on biomass production (*Figure 3a*). Specifically, microbial diversity reduced the temporal variation in biomass production, thus increasing its stability. In turn, greater stability in biomass production reduced the temporal variation in soil carbon assimilation, thus stabilizing soil carbon assimilation (*Figure 3b*).

To identify microbial taxa that may influence an ecosystem function at a given time, we used a randomization approach (*Mori et al., 2016*; *Gotelli et al., 2011*) to calculate the standardized effect size (SES) of each fungal and bacterial taxon on each ecosystem function. This method generates a distribution for the null hypothesis that the effect of the presence of a taxon on a given ecosystem function at any given time occurs by chance. The influence of a taxon on an ecosystem service at a given time was considered to be significant when the effect of the presence of the taxon on an ecosystem function was greater than 1.96 (the 95 percentile of the t-distribution) times the SD of the

null distribution. As we considered more time points, we detected a greater proportion of microbial taxa that were positively, or negatively, related to supporting an ecosystem function as more time points were considered (*Figure 4a*, see *Supplementary file 2* for a summary for their taxonomic assignment). In addition, we found that the proportion of soil taxa with positive or negative associations with ecosystem functioning increased as more functions as well as more time points were independently considered (*Figure 4*), demonstrating that different microbial taxa are important for different ecosystem functions at different times. These relationships were found for bacterial and fungal taxa combined (*Figure 4a,b*), as well as for bacterial (*Figure 4c,e*) and fungal taxa (*Figure 4d,*

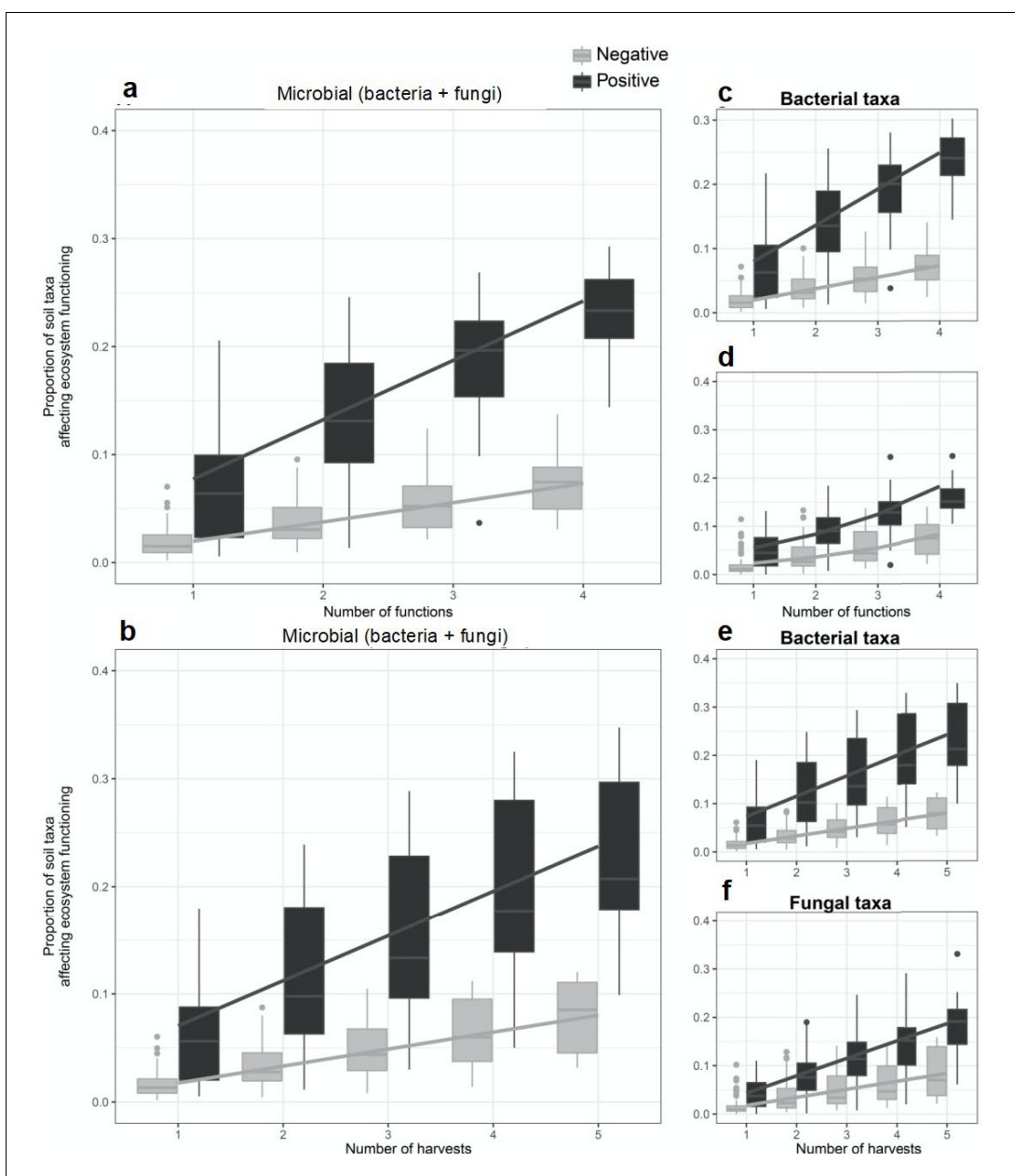

**Figure 4.** The proportion of soil microbes that support ecosystem functions increases with the number of functions and time points considered. A higher proportion of soil taxa affected ecosystem functioning, both positively (black bars) and negatively (grey bars), when more ecosystem functions (**a, c, d**) and number of time points (**b, e, f**) were independently considered. This result was found for the scaled average of fungal and bacterial taxa (**a, b**), the soil bacterial (**c, e**), and soil fungal taxa (**d, f**). Regression lines indicate generalized linear model fits and box plots summarize observed data.

*f*) considered separately. Moreover, we found a larger cumulative proportion of soil taxa with positive than negative effects on ecosystem functioning with increasing number of functions and times considered (black versus grey lines in *Figure 4*).

Of the microbes that were found to have a significant positive association with a function at any given time, we found that the average temporal stability in the abundance of these fungi and bacteria (species stability $CV_{species}^{-1}$) was positively related to the temporal stability of that function, with the exception of soil carbon assimilation (*Figure 5*). In other words, the more stable the abundance of the taxa that support an ecosystem function at any time, the greater the stability of that ecosystem function (see *Supplementary file 1*–tables 9 and 10 for the taxonomic assignment of the taxa that were significantly related to functions at more than one time point). These species stability-ecosystem function stability relationships could be explained by the most extreme soil biodiversity level (the 'sterile treatment', *Supplementary file 1* – table 8). We also found that for all four ecosystem functions, a lower temporal synchrony among those taxa that had a positive association with a given function at any given time point also significantly related to a greater stability of that ecosystem function (*Figure 5*). These species asynchrony-ecosystem function stability relationships were largely independent of the most extreme sterile soil biodiversity level. After first accounting for the sterile soil biodiversity level, the residual effect of microbial asynchrony on plant diversity and litter decomposition stability remained significant (p=0.036 and 0.048, respectively) and the effect on litter decomposition stability was marginally non-significant (p=0.058, *Supplementary file 1* – table 9).

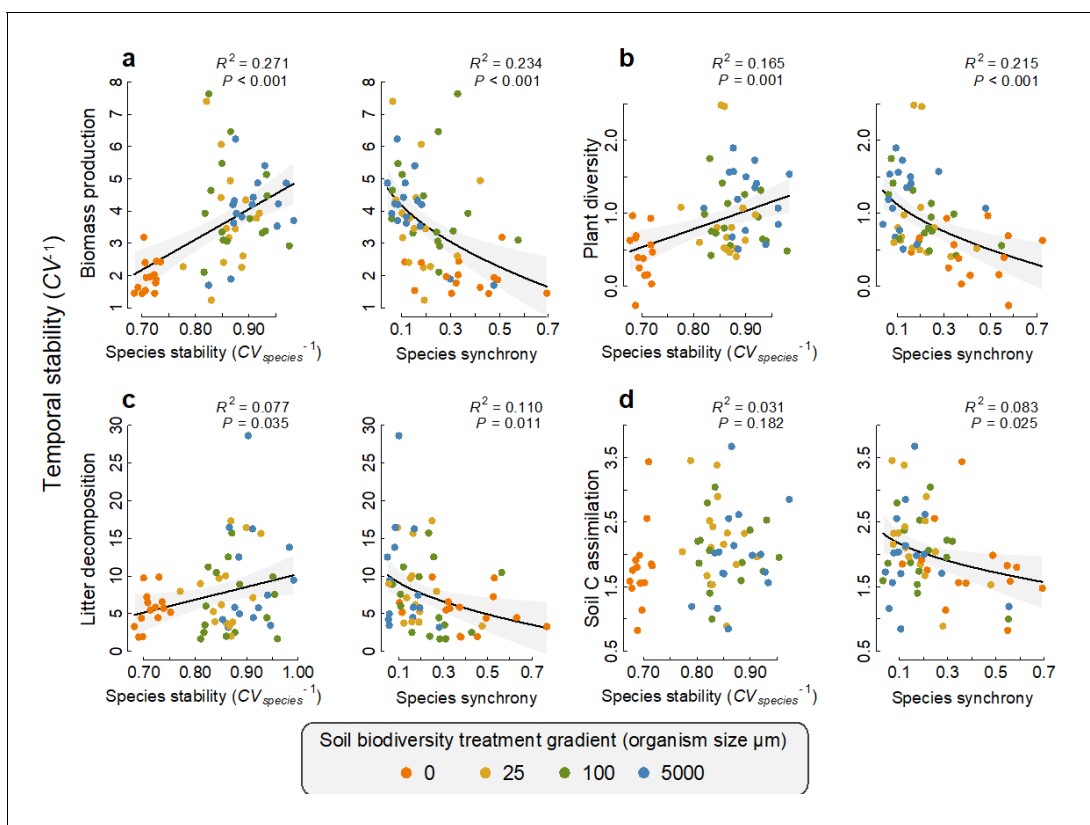

**Figure 5.** Relationships between the temporal stability of each ecosystem function with species temporal stability and synchrony. (**a**) Temporal stability of plant biomass production, (**b**) temporal stability of plant diversity, (**c**) temporal stability of litter decomposition, and (**d**) temporal stability of soil carbon assimilation are shown in relation to the average stability in the abundance of the fungi and bacterial taxa (species stability) and the temporal synchrony among fungi and bacteria that were found to support each function. The relationships for fungi and bacteria are shown separately in *figure 1*. Regression lines and 95% confidence bands are shown for significant relationships (p<0.05).

The online version of this article includes the following figure supplement(s) for figure 5:

**Figure supplement 1.** Temporal stability in ecosystem functions relationships with the temporal stability and synchrony in the abundance of individual fungal and bacterial taxa.

## Discussion

Here, we demonstrate that that the maintenance of greater soil microbial diversity not only supports ecosystem functions but also stabilizes multiple ecosystem functions simultaneously. This is supported by our analysis of individual functions, the average response in stability of all four functions (*Maestre et al., 2012*), and the multiple threshold approach (*Byrnes, 2014*). Our results show that both fungal and bacterial richness support greater stability in multiple functions simultaneously with the strongest effect of microbial richness on stability occurring when functions need to achieve at least 70% of their maximum observed stability. The effect of soil biodiversity loss on multifunctional stability was strongest in our most extreme 'sterile' treatment, suggesting a loss of microbial richness over 50% as a potential tipping point over which the stabilizing effect of soil microbial diversity might drop abruptly.

We found that the stabilizing effect of a diverse soil microbial community is attributed to the asynchronous temporal dynamics in the soil microbial community composition. Despite the strong contribution of the sterile soil treatment to the results, we still found the effects of the composition and temporal asynchrony to be important for stabilizing ecosystem functions even after accounting for our most extreme treatment. To our knowledge, our results demonstrate for the first time that the maintenance of a more diverse soil microbial community allows for greater asynchronous temporal dynamics in the microbial communities to result in the maintenance of higher nutrient and carbon cycling processes through time (reflected here by four key ecosystem functions). This implies that different microbes can complement each other by functioning at different times to provide a stable support for ecosystem functions.

Asynchrony in microbial communities may arise as microbial community compositional changes track the phenology of the associated plant communities through plant nutrient uptake, root exudates, and litter inputs that generate changes in soil chemistry and resources for soil microbes (*Marschner et al., 2002*; *Bray et al., 2012*; *Buscardo et al., 2018*; *Upton and Hofmockel, 2019*). For instance, different N-cycling microbes exhibit temporal seasonal dynamics as they are active at different times to contribute to nitrification and denitrification processes that are also linked to plant community development (*Prosser and Nicol, 2012*; *Regan, 2017*). Additionally, microbial community composition changes during decomposition as microbes have different resource acquisition strategies where some fast-cycling microbes acquire readily available carbon early, while slow-cycling microbes decompose recalcitrant carbon at later stages (*Herzog, 2019*; *Alavoine and Bertrand, 2020*; *Su et al., 2020*). Thus, by having a greater diversity of microbes there is a greater opportunity for different microbes to perform specific functions associated with the different stages of litter decomposition, nutrient and carbon cycling that together maintain the broad ecosystem functions of plant productivity, diversity, decomposition, and carbon assimilation through time.

Past studies assessing soil microbial diversity-ecosystem functioning relationships have shown that the suppression of soil microbial diversity and abundance can have little impact on particular functions measured at a single time point, possibly due to high redundancy among taxa (*Griffiths et al., 2000*; *Fitter et al., 2005*; *Allison and Martiny, 2008*; *Louca et al., 2018*). However, here we show that as more functions and time points are considered, a greater proportion of the microbial community is required to support these functions and correspondingly redundancy, or the proportion of microbes not associated with ecosystem functions, fades away. This is supported by the consistent trend that a greater number of taxa were found to positively support ecosystem functions when more functions and time points were considered, highlighting that increasing the number of taxa ensures the maintenance of the functioning of the overall ecosystem over time (*Isbell, 2011*; *Hautier, 2015*; *Yachi and Loreau, 1999*). This suggests that whereas a large number of soil microbial taxa may overlap in supporting a single function at a single time point, fewer and fewer taxa may overlap in supporting multiple functions over multiple time points. Similar to studies assessing the functional redundancy of plant communities (*Hautier et al., 2018*; *Isbell, 2011*), as more ecosystem functions and times were considered in the present study, more soil taxa were found to affect the overall functioning of the ecosystem. While high soil microbial diversity may appear functionally redundant for a given broad function, such as microbial respiration and decomposition (*Schimel and Schaeffer, 2012*), there are studies demonstrating low functional redundancy in soils in both broad functions (soil respiration) and specialized functions (degradation of specific organic compounds such as cellulose and microbial toxins)(*Wohl et al., 2004*; *DelgadoBaquerizo, 2016*). Here we

found, for instance, that the mycorrhizal fungi *Diversispora* was frequently associated with positive effects on biomass production, but not other functions, while the plant pathogen *Fusarium* was frequently found to positively influence plant diversity and no other functions (see *Supplementary file 2* – table 2). Such findings are logical as mycorrhiza are well known to drive plant biomass production (e.g. *van der Heijden et al., 2008*) and pathogens can maintain plant diversity (*Petermann et al., 2008*; *Schnitzer et al., 2011*).

It is important to note that the composition and diversity of other key groups of soil biota, including protozoa and nematodes not assessed in our study were likely also affected by the sieving treatments and may also have contributed to the observed effect of our soil biodiversity measures on ecosystem functioning and stability either directly or through trophic interactions with soil microbes (*de Vries and Shade, 2013*; *Bradford, 2002*; *Trap et al., 2016*; *Thakur and Geisen, 2019*). Assuming this was the case, this would suggest that the broader spectrum of soil biodiversity beyond the soil microbiome may also hold a key role in promoting multiple ecosystem functions (*Delgado-Baquerizo et al., 2016*; *Bradford, 2014*; *Wagg et al., 2014*; *Jing et al., 2015*; *Delgado-Baquerizo et al., 2020*; *Mori et al., 2016*; *Wagg et al., 2019*), their temporal stability and by extension possibly their resistance and resilience to environmental perturbation (*Griffiths et al., 2000*; *Delgado-Baquerizo et al., 2017*; *Griffiths and Philippot, 2013*). Nevertheless, the effects of soil biodiversity loss and the role of functional redundancy in the soil microbiome for stabilizing soil functions have yet to be fully elucidated in the context of the complexity of interactions among the various trophic levels of soil organisms (*Soliveres et al., 2016*).

Studies focussing on plant communities have demonstrated the importance of plant diversity for promoting biomass production (*Cardinale, 2012*; *Hooper et al., 2012*) and the temporal stability in biomass production (*Tilman and Downing, 1994*; *Hector and Bagchi, 2007*; *Hautier et al., 2018*), as well as building soil carbon pools (*Lange et al., 2015*; *Yang et al., 2019*). In our model system, we were able to reveal that soil microbial diversity had a direct positive effect on the stability of biomass production that in turn stabilized soil carbon assimilation. Specifically, soil microbial diversity supported a higher stability in biomass production due to its strong effect on lowering the temporal variation in plant biomass production, which was independent of plant diversity. While experimental manipulations of plant diversity are well known to enhance plant productivity (*Tilman and Downing, 1994*; *Cardinale, 2012*; *Hooper et al., 2012*; *Tilman et al., 2006*; *Hector et al., 2010*; *Isbell, 2011*; *Hautier, 2015*), the lack of a direct effect of plant diversity on the stability of biomass production and carbon in our experiment may be not surprising, because all our treatments started off with the same plant diversity while soil communities were experimentally manipulated. At the lowest levels of soil biodiversity, we found that plant productivity initially increased over time and was the most productive. This was driven by the rapid growth of *L. perenne* grass, which is highly competitive for soil nitrogen and is a plant species that is hardly affected by changes in arbuscular mycorrhizal diversity when grown alone (*Wagg et al., 2011a*; *Wagg et al., 2011b*). Its initial high productivity may be due to initial release from pathogens and plant competition at low levels of soil biodiversity. As time progressed, the productivity of this highly productive grass was unsustainable, and the productivity crashed compared with our more diverse treatments, resulting in the 'boom and bust' temporal trend. We attribute this decline in productivity at the lowest level of soil biodiversity to the inability of the soil community to maintain the presence and growth of legumes and impaired nutrient availability and reduced nutrient cycling processes needed to maintain higher plant community productivity (*Wagg et al., 2019*; *Roscher, 2013*; *Marquard, 2009*; *Schmidtke et al., 2010*).

In contrast to the stability in biomass production, we found that the effect of soil microbial diversity on stabilizing soil carbon assimilation was indirect through its effect on promoting the stability in plant biomass production. This is logical since the primary path for atmospheric carbon to become assimilated into soils is through photosynthesis, upon which plant biomass production is based, that is then transferred to soil microbes symbiotically or through the microbial accumulation of plant-derived carbon through decomposition processes (*De Deyn et al., 2008*; *De Deyn et al., 2011*; *Merino et al., 2015*). Together with our findings, these results provide support for the concept that there is a tight link between the diversity and composition of the soil microbiome belowground and the biomass production of plants aboveground that together promote and stabilize soil carbon assimilation.

Earlier studies showed that the diversity and composition of the soil microbiome has a big impact on ecosystem functioning. Yet it had not been shown previously whether altered temporal variations

in community composition and biodiversity in the soil microbiome impairs the stability of soil functions. Our study provides a first proof of concept that severe degradation of belowground biological communities reduces multiple ecosystem functions and their stability over time with a likely tipping point (>50% microbial biodiversity loss). Moreover, our work highlights that asynchrony of different microbes can improve the stability of ecosystem functioning over time. Our result is concerning as soil biodiversity is being lost in many parts of the world in response to anthropogenic disturbances and increasing land-use intensification compounded by effects of climate change (*Helgason et al., 1998*; *Birkhofer, 2008*; *Verbruggen, 2010*; *Tsiafouli et al., 2015*; *Stavi and Lal, 2015*). Such biological degradation of soils may likely have cascading effects on their ability to maintain diverse productive plant communities that promote carbon sequestration. Our study shows that this is because there are fewer microbes that are needed to support the cycling of soil nutrients and plant growth at different times throughout the development and growth of plant communities. Consequently, anthropogenic activities that simplify soil diversity are likely to diminish the reliable provisioning of essential ecosystem services.

## Materials and methods

### Experimental design

Experimental grassland communities were established in self-contained mesocosms described in detail elsewhere (*Wagg et al., 2014*; *van der Heijden et al., 2016*). Briefly, mesocosms were 23 cm in diameter and 34 cm in height with incoming air and water passing through hydrophobic (0.2 μm pore size) and hydrophilic (0.22 μm pore size) sterilizing filters (all Millex-FG$_{50}$; Millipore Corporation, Billerica, MA) to prevent microbial contamination. All mesocosm components were sterilized by autoclaving at 120℃ for a minimum of 20 min, with the exception of the Plexiglas tops and the PVC bottoms, which were sterilized by a 20 min submersion in 0.5% hypochlorite followed by a 20 min submersion in 70% ethanol with Tween 20 and placed in the laminar-flow hood, under which all soil and plant communities were assembled and harvested as described below. The bottom of each mesocosm was covered with a 1 cm layer of quartz stones (roughly 1 cm$^3$ in size) and topped with a propyltex screen (0.5 mm mesh size, Sefar AG, Heiden, Switzerland). Mesocosms were then filled with 5.5 kg (dry mass) of a 50/50 field soil-to-quartz sand mix that was previously sieved through a 5 mm mesh sterilized by autoclaving (120℃ for 90 min). The soil for this substrate came from a grassland field in Zürich, Switzerland (47° 25′ 38.71″ N, 8° 31′ 3.91″ E).

### Soil inoculum

Soils were collected from three different fields, having different soil histories. Two of the fields were located in Therwil, Switzerland (47° 30′ 8.9964″ N, 7° 32′ 21.8292″ E, one managed with organic fertilizer (soil attributes: pH = 7.9, P=50.3 mg/kg, N = 41.8 mg/kg, and K = 1.9 mg/kg) and the other with mineral fertilizer (soil attributes: pH = 7.4, P=47.5 mg/kg, N = 44.4 mg/kg, and K = 2.0 mg/kg). A third field as used to collect soil from a site in Freiburg, Germany (47° 58′ 26.058″ N, 7° 46′ 31.5336 E, soil attributes: pH = 7.4, P=41.1 mg/kg, N = 44.8 mg/kg, and K = 1.7 mg/kg). At each site, soil cores (size 10 cm deep) were collected every meter along four transects that were 4 m apart and homogenized per site by sieving through a 5 mm sieve. Different soil origins were used to block for potential site-specific effects so that general conclusions on the soil biodiversity and community compositional treatments can be inferred. The soil biodiversity gradient then was generated by sieving the soils to different sizes (*Bradford, 2002*; *Wagg et al., 2014*). Here we used four sieving treatments, 5000 μm (5 mm), 100 μm, 25 μm, and 0 μm (sterilized soil inoculum). Soil material not passing through the sieves was autoclaved and included with the unsterilized sieved portion that was used to inoculate the mesocosm substrate. Each mesocosm received inoculum of one of the four sieved soil inoculum treatments. The inoculum which consisted of ≈ 4.5% of the total substrate volume was then thoroughly mixed with the mesocosm substrate. Each of the four soil inoculum treatments was replicated five times using soil from each of the three soil histories for a total of 60 mesocosms (four inoculum treatments by three inoculum origins by five replicates). The mesocosms were set up over eight days and all subsequent data collection methods followed the order in which the mesocosms were set up. The day on which each was set up was used as a blocking factor in the subsequent analysis. Mesocosms were planted with a plant community of 34 individuals: 14 grasses

(12 *Lolium perenne* and 2 *Festuca pratensis*), 14 legumes (12 *Trifolium pratense* and 2 *Lotus corniculatus*), and 6 forbs (2 *Achillea millefolium,* 2 *Plantago lanceolate,* and 2 *Prunella vulgaris*). Seeds of each species were surface sterilized in 2.5% hyposodium chlorate for five minutes and rinsing in sterile $H_2O$. Seeds were germinated on 1% Agar in Petri dishes and the timing was staggered so that each species exhibited the presence of cotyledon(s) or radicle when planted. Microcosms were established in a glasshouse with natural light subsidized by 400 W high-pressure sodium lamps in order to maintain an environment of 16 hr/25°C days and 8 hr/16°C nights with a light level above 300 W/m². Soil moisture in the microcosms was maintained by watering twice weekly with $dH_2O$ that first passed through a sterilizing filter before entering the microcosm. Since greenhouse conditions maintain a constant environment, which may not reflect natural temporal environmental variations, we varied the watering regime to simulate an extended period without rain. The variation in precipitation was applied to all of the experimental communities at the same time by withholding watering for 10 days beginning five-and-a-half weeks before each harvest.

## Ecosystem functions

Plant diversity (Shannon *H'*), plant productivity (aboveground biomass produced between harvests), litter decomposition, and carbon assimilation were measured every 11 weeks over 55 weeks (five harvest dates). Plant diversity and plant productivity were measured by harvesting plants 5 cm above the soil surface and dried at 65°C for a minimum of 72 hr. Note that this harvesting treatment mimics typical cutting regimes in real grassland. Litter decomposition was measured in 0.5 mm propyltex mesh litterbags (6 × 6 cm) filled with 1 g of dried *Lolium multiflorum* shoots that were sterilized by autoclaving. Litterbags were buried just below the surface of the soil substrate in each mesocosm. At each harvest, the litterbag was removed, washed clean of soil, dried at 65°C and the remaining litter mass was weighed. The amount of the initial 1 g of litter lost (in mg lost per 11 weeks) was calculated as decomposition. A new litterbag was inserted every harvest. Carbon assimilation was quantified by injecting each mesocosm with 40 ml of ¹³C-labelled $CO_2$ (99%) gas at 36 and 18 hr before each harvest. At each harvest, six soil cores (1.7 mm diameter) were taken to the depth of the mesocosm (~20 g of fresh soil). Soil cores were homogenized and a 0.5 g subsample was frozen at –20°C for DNA extraction. The remaining soil sample was lyophilized and analyzed for ¹³C content using a Delta Plus XP isotope ratio mass spectrometer (Thermo Finnigan). The ¹³C ratio was calculated in relation to the international standard VPDB (Vienna Pee Dee Belemnite). The more negative the ¹³C ratio, the less carbon was fixed.

## Soil community quantification

Here, we focused on soil bacterial and fungal communities as they are the most conspicuous and diverse components of the soil microbiome and they play a major role in the functional organization of ecosystems. Following each harvest, DNA was extracted from 500 mg of the homogenized soil samples taken from each microcosm mentioned above. Extraction was done using the FastDNA SPIN Kits for Soil (MP Biomedicals, Switzerland). Using a Quant-iT PicoGreen (Molecular Probes, Eugene, OR) on a luminescence spectrometer (Perkin Elmer, LS 30, Rotkreuz Switzerland), the extracted DNA was quantified. DNA concentrations were determined using a Qubit fluorometer (Life Technologies, Paisley, UK). A barcoded high-throughput sequencing approach was employed to assess the diversity and composition of bacterial and fungal communities. Bacterial communities were examined by amplifying the V3-V4 region of the bacterial 16S rRNA gene using the primers 341F (CCTACGGGNGGCWGCAG) and 805R (GACTACHVGGGTATCTAATCC) (*Herlemann, 2011*). Fungal communities were examined by amplifying the ITS (Internal Transcribed Spacer) region using the primers ITS1F (CTTGGTCATTTAGAGGAAGTAA) (*Gardes and Bruns, 1993*) and ITS2 (GCTGCG TTCTTCATCGATGC) (*White et al., 1990*) targeting the ITS1 region. Each primer was tagged with a 5-nucleotide-long padding sequence and an 8-nucleotide-long barcode. PCR (Polymerase Chain Reaction) was conducted on a Biorad PCR Instrument (Biorad, Hamburg, Germany) using the 5PRIME HotMaster Taq DNA Polymerase (Quantabio, Beverly, MA) in 20 μl of reaction mixture. To alleviate stochastic PCR effects of individual reactions, PCR (Polymerase Chain Reaction)s were performed in triplicate for each DNA sample. Thermal cycling conditions for bacterial 16S rRNA were 2 min initial denaturation at 94°C followed by 30 cycles of 30 s denaturation at 94°C, 30 s annealing at 53°C and 30 s elongation at 65°C, and finally a elongation of 10 min at 65°C. Thermal cycling

conditions for fungal ITS (Internal Transcribed Spacer) comprised a 2 min of initial denaturation at 94℃ followed by 30 cycles of 45 s denaturation at 94℃, 1 min annealing at 50℃ and 90 s elongation at 72℃, and a final elongation of 10 min at 72℃. Amplicons were loaded on a 1% agarose gel to examine PCR (Polymerase Chain Reaction) efficiency and the lack of PCR amplicons in non-template control reactions. After PCR (Polymerase Chain Reaction), replicates were pooled for each sample and the concentration of amplicon DNA was determined using PicoGreen (Molecular Probes Inc, Eugene, OR) on the Varian Turbo GTI fluorescence plate reader (Varian Inc, Poalo, CA). Two amplicon libraries (~2 µg each) were assembled by combining barcoded DNA samples and purifying twice with the Agencourt AMPure XP PCR (Polymerase Chain Reaction) Purification system (Beckman Coulter, IN). For each library, a final volume of 100 µl was obtained by eluting in sterile miliQ water.

For all amplicons, Illumina 300 bp paired-end sequencing was performed at the Functional Genomics Centre of Zürich (http://www.fgcz.ch). For bacterial and archaeal 16S rRNA genes, the quality of R1 and R2 reads was determined using FastQC (*Andrews, 2010*). Reads were trimmed to remove base pairs from the end of reads after read quality per sample declined (25 and 50 bp for read1 and read2, respectively). Reads were then merged, allowing a minimum overlap of 15 bp using FLASH v1.2.11 (*Magoč and Salzberg, 2011*). FASTA format sequences were extracted from FASTQ files quality filtered using the PRINSEQ-lite v0.20.4 (*Schmieder and Edwards, 2011*). Filtering parameters were: GC range 30–70, minimum mean quality score of 20, no ambiguous nucleotides, low sequence complexity filter with a threshold of 30 in the DUST algorithm. In a next step, the reads were demultiplexed using an in silico PCR (Polymerase Chain Reaction) approach as part of usearch (v11), allowing max one mismatch in the barcode-primer sequence but not at the 3-prime ends. Sequences were then clustered into OTUs, based on 97% similarity, using the UPARSE pipeline (*Edgar, 2013*). Taxonomical information was predicted for bacterial and fungal OTUs based on the SILVA v128 (*Pruesse, 2007*) and UNITE (V7.2) (*Kõljalg, 2005*), respectively. For fungi, the taxonomic prediction was verified using ITSx (*Bengtsson-Palme, 2013*) and non-fungal OTUs were excluded.

## Microbial richness-multifunctional stability relationships

All statistical analyses and data manipulations were done using R software (version 3.0.0), including the packages 'vegan' and 'nlme'. Mixed-effect models assessing the effects of our soil biodiversity treatment gradient and sampling time points on microbial richness and each ecosystem function were fitted using the packages 'asreml' (VSN International Ltd.) and 'pascal' (available at https://github.com/pascal-niklaus/pascal; *Wagg, 2021*; copy archived at swh:1:rev:4759fc326603fe70-be1f49c17e7c8227f37766af) and included the identity of the mesocosm and soil inoculum origin within blocks as random terms as well as the autocorrelation of residuals across sequential harvests. The treatment gradient and sampling time were both fit as continuous variables to trends along our gradient and through time and were followed by fitting the treatment gradient and time as factors to assess non-linearity in the relationships with our experimental gradient and time. Where ecosystem functions were found to follow non-linear temporal trends, polynomial regression (from cubic to fourth order) was used to fit non-linear relationships and the best fit was assessed using the marginal $R^2$ of the model and AICc.

The temporal stability for each ecosystem function (plant productivity, plant diversity, litter decomposition and carbon assimilation) within mesocosms was calculated as the inverse of the coefficient of variation: the ratio of the temporal mean of a function to its temporal SD over the five time points (*Tilman et al., 2006*; *Pimm, 1984*; *Tilman, 1999*). The stability of each ecosystem function was assessed separately for its relationship with microbial richness (the average richness of fungi and bacteria per mesocosm over all time points) using a mixed-effect regression model with soil inoculum origin within experimental blocks as a random term. We also included a contrast term to test whether our microbial diversity-stability relationships were due to the most extreme (sterile) soil biodiversity treatment. In the same way, we assessed the relationship between the average stability of multiple functions after first scaling each function between 0 and 1 (multifunctional stability) with fungal and bacterial richness and a soil microbial diversity index, which we calculated by averaging the fungal and bacterial richness after first scaling them between 0 and 1 for each time point.

We also used the multifunctional threshold method (*Byrnes, 2014*) to quantify the number of functions with temporal stability exceeding a given threshold, where thresholds are varied along a gradient from 5% to 95% of the maximum observed stability of the function. This method allows us to determine whether our measures of soil microbial diversity support multiple functions at high

levels (*Byrnes, 2014*; *Pasari et al., 2013*). Here, we explored threshold values between 5% and 95% at 1% intervals. We examined the relationships of soil bacterial and fungal richness as well as their microbial diversity index with the number of functions above a threshold by fitting generalized linear mixed-effects models (GLMMs) with a negative binomial distribution with a logit link function and soil inoculum origin and block as random effects. Separate models were fitted for each of the threshold levels and the slope and associated 95% confidence intervals were recorded.

Since greater plant diversity is also known to enhance biomass production and soil carbon assimilation (*Tilman and Downing, 1994*; *Hector and Bagchi, 2007*; *De Deyn et al., 2011*; *Zavaleta et al., 2010*; *Lange et al., 2015*; *Hautier et al., 2018*; *Yang et al., 2019*), we assessed the indirect effects of soil microbial diversity on the stability of biomass production and soil carbon assimilation through their influence on plant diversity using a multi-model comparison and SEM) approach. First, we assessed all possible combinations of soil microbial diversity and plant diversity as predictors of the stability of plant biomass production as well as all possible combinations of soil microbial diversity, plant diversity, plant biomass production and the stability in plant biomass production as predictors of the stability in soil carbon assimilation using linear regression with soil inoculum origin and block as random effects. Models were compared based on AICc and marginal $R^2$ values. We then used the results of these models to construct SEM models to assess direct versus indirect effects of soil microbial diversity on the stability of biomass production and soil C assimilation. Since assessing direct and indirect paths of microbial diversity on the stability of biomass production and carbon assimilation is a fully saturated model and since stability is the ratio of the temporal mean to variance (SD), we decomposed the effects of soil microbial diversity on stability via their effects on the temporal mean and SD of biomass production and soil C assimilation. This also allows to assess the relative effects of whether stability is driven more by a higher overall value (temporal mean) or smaller temporal variation or both relatively equally.

## Effects of microbial taxa on different functions at particular times

We tested whether the accumulation of taxa affected ecosystem functioning across functions and times. In order to do so, we sampled all combinations of the four functions at each time and all combinations of times for each function considered and quantified the proportion of unique taxa that promoted or reduced ecosystem functioning at least once. The effect of each operational taxonomic unit (taxon) on a given ecosystem function was generated by randomly reassigning the ecosystem function and calculating the difference in the mean ecosystem function when the taxon was absent or present over 999 iterations. This generated simulated means and SDs. The taxon-specific standardized effect size (SES) was then determined as the difference in the observed and simulated effects divided by the simulated SD (*Mori et al., 2016*; *Gotelli et al., 2011*). Taxa were considered to have a significant impact on an ecosystem function if their |SES| >1.96, corresponding to a 5% error probability that the difference was not zero. We then examined how the composition of soil bacterial and fungal taxa that were significantly associated to greater or reduced ecosystem functioning changed with the number of functions or times considered using GLMMs with beta-binomial distribution and soil inoculum origin as random effect. If completely unique sets of taxa affect different functions at any particular time or any particular function at different times, then the relationship between the proportion of taxa affecting ecosystem functioning and the number of functions or times considered, respectively, would be a positive linear relationship with slope 1. On the other hand, if completely identical sets of taxa were involved, then the relationships would be horizontal with slope 0 (*Hector and Bagchi, 2007*; *Isbell, 2011*).

Using the taxa that were found to be significantly positively related to an ecosystem function at any given time, we then calculated their stability (here termed species stability) and synchrony in their abundance. Species stability is the average coefficient of variation in the abundance of each fungal and bacterial taxon. Synchrony was calculated as the average of the covariance among all pairs of fungal and bacterial taxa after first standardizing their abundance (mean = 0, SE (Standard Error) = 1). We then assessed the relationships between the stability and synchrony of microbial OTUs that positively supported a function at any given time with the temporal stability of that function using separate regression models and contrasts assessing the influence of the sterile soil treatment on the relationship as was done for the richness-stability relationships mentioned above.

Code availability: R code will be made publicly available upon acceptance.

Data availability: All data will be made publicly available upon acceptance.

## Acknowledgements

We are grateful to Klaus Schitterer, Ernst Brack, Jochen Mayer, and Paul Mäder at the DOK Trial for soil collection and access to the field site, and Jan-Hendrik Dudenhöffer, Christoph Sax, and Alain Held for greenhouse support. This project was supported by a grant from the Swiss National Science Foundation (SNF grant 137136) awarded to MvdH and BS. CW was supported by the German Research Foundation grant FOR 456/FOR 1451 as part of the Jena Biodiversity Experiment.

## Additional information

### Competing interests

Bernhard Schmid: Reviewing editor, *eLife*. The other authors declare that no competing interests exist.

### Funding

| Funder | Grant reference number | Author |
|---|---|---|
| Schweizerischer Nationalfonds zur Förderung der Wissenschaftlichen Forschung | 137136 | Marcel van der Heijden |
| Deutsche Forschungsgemeinschaft | FOR 1451 | Bernhard Schmid |

The funders had no role in study design, data collection and interpretation, or the decision to submit the work for publication.

### Author contributions

Cameron Wagg, Conceptualization, Data curation, Formal analysis, Supervision, Visualization, Writing - original draft, Writing - review and editing; Yann Hautier, Conceptualization, Formal analysis, Methodology, Writing - original draft, Writing - review and editing; Sarah Pellkofer, Data curation, Investigation, Methodology, Writing - review and editing; Samiran Banerjee, Formal analysis, Methodology, Writing - review and editing; Bernhard Schmid, Supervision, Funding acquisition, Writing - review and editing; Marcel GA van der Heijden, Conceptualization, Supervision, Funding acquisition, Writing - review and editing

### Author ORCIDs

Cameron Wagg (iD) https://orcid.org/0000-0002-9738-6901
Sarah Pellkofer (iD) https://orcid.org/0000-0003-0208-0350
Bernhard Schmid (iD) https://orcid.org/0000-0002-8430-3214

### Decision letter and Author response

Decision letter https://doi.org/10.7554/eLife.62813.sa1
Author response https://doi.org/10.7554/eLife.62813.sa2

## Additional files

### Supplementary files

• Supplementary file 1. Results of ANOVA models assessing the effects of soil biodiversity treatments on microbial richness, composition and ecosystem functions, and effect of microbial richness, asynchrony, and stability in the abundance of microbes on the stability of ecosystem functions.

• Supplementary file 2. Summary results for the taxonomic assignment of operational taxonomic units (OTUs) having a significant positive and negative association with an ecosystem function at each time point.

• Transparent reporting form

## Data availability

All data generated or analysed during this study are available online at: https://figshare.com/s/790d5ba1fb7f8e670a27.

The following dataset was generated:

| Author(s) | Year | Dataset title | Dataset URL | Database and Identifier |
|-----------|------|---------------|-------------|-------------------------|
| Cameron W | 2020 | Data: Diversity and asynchrony in soil microbial communities stabilizes ecosystem functioning | https://figshare.com/s/790d5ba1fb7f8e670a27 | figshare, 10.6084/m9.figshare.12982937 |

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
