## [Decision Letter]

**Acceptance summary:**

This study is one of the first to link soil biodiversity to ecosystem stability, and as such, it has potential to become a reference paper in the field. The work extends our understanding of diversity effects on ecosystem function beyond plants, the focus of most contributions in this area.

**Decision letter after peer review:**

Thank you for submitting your article "Diversity and asynchrony in soil microbial communities stabilizes ecosystem functioning" for consideration by *eLife*. Your article has been reviewed by two peer reviewers, and the evaluation has been overseen by Detlef Weigel as the Senior and Reviewing Editor. The following individual involved in review of your submission has agreed to reveal their identity: Akira Mori (Reviewer #2).

The reviewers have discussed the reviews with one another and the Reviewing Editor has drafted this decision to help you prepare a revised submission.

Summary:

This well-written paper concerns a timely topic: What is the relationship between microbial diversity/asynchrony and ecosystem stability? The study is one of the first to link soil biodiversity to ecosystem stability, and as such, it has potential to become a reference paper in the field. The authors quantified diversity and functions with several rigorous approaches, and the findings are overall sound, providing important empirical data that have often been lacking for soil – an important compartment of ecological systems. Importantly, the work extends our understanding of diversity effects on ecosystem function beyond plants, the focus of most contributions in this area.

The work, however, would considerably benefit from more appropriate framing of the results. In the current form, the emphasis is on stabilizing effects and asynchronous dynamics of soil microbes, claimed to ensure multiple ecosystem functions, but with little attention to possible processes and mechanisms underlying these observations. Not only must there be a discussion of potential processes and mechanisms for broader appeal of the work, and to advance the theory of biodiversity-ecosystem functioning relationships, but from the outset (that is, in the Introduction) there is already a need to improve the framing of the study. We acknowledge that it is not fully possible to identify possible mechanisms based on the study design, but the area in general has to begin to go beyond merely observing relationships between diversity and function(s).

Major comments:

State already in the Abstract what members of the microbial community are considered.

You state "negatively affects multiple ecosystem functions". This is too vague. Please be more specific and concrete what you mean.

Please explain what exactly is meant with "making biodiversity a cornerstone for long-term sustainability". The stabilizing effects of biodiversity to ensure stable provisioning of ecosystem services are important. However, the current expression needs prior knowledge, which a general audience is unlikely to have. Sustainability itself is used very widely. Please be more specific here and elsewhere in the text.

"Functions" is also used without clear definitions. It is helpful for readers if clear and brief definitions are provided for this terminology. Without this, it is hard for most readers to follow the arguments in the second paragraph.

What is meant with "soil degradation"? It can be loss of soil biota, but other possibilities can be also included, such as pollution, erosion and so on.

Please provide assumptions that motivate the experimental manipulation, e.g., how this diversity loss is realistic reflecting a scenario of soil degradation. Best to do this early on, for example at the end of the Introduction.

Define stability and state how it was calculated.

Define temporal asynchrony, and "asynchronous dynamics". Because of such dynamics, microbes are less redundant. We do agree that this is important, but the argument here sounds more like interpretation.

---

## [Author Response]

Major comments:State already in the Abstract what members of the microbial community are considered.

We now state that we assessed fungal and bacterial communities: “Here we experimentally quantified the contribution diversity and the temporal dynamics in the composition of soil fungal and bacterial communities.”

You state "negatively affects multiple ecosystem functions". This is too vague. Please be more specific and concrete what you mean.

This sentence has now been re-written for clarity to: “For example, evidence is mounting that the reduction in species diversity can reduce multiple ecosystem functions, such as nutrient and carbon cycling processes and plant productivity.”

Please explain what exactly is meant with "making biodiversity a cornerstone for long-term sustainability". The stabilizing effects of biodiversity to ensure stable provisioning of ecosystem services are important. However, the current expression needs prior knowledge, which a general audience is unlikely to have. Sustainability itself is used very widely. Please be more specific here and elsewhere in the text.

This sentence has now been re-written for clarity to: “Reduced biodiversity has also been shown to destabilize ecosystem functioning over time, and thus the maintenance biodiversity can be key to long-term ecosystem sustainability.”

"Functions" is also used without clear definitions. It is helpful for readers if clear and brief definitions are provided for this terminology. Without this, it is hard for most readers to follow the arguments in the second paragraph.

Functions has now been defined up front with the revision of the text where we now state: “ecosystem functions, such as processes that drive nutrient and carbon cycling and plant productivity.”

What is meant with "soil degradation"? It can be loss of soil biota, but other possibilities can be also included, such as pollution, erosion and so on.

We have added: “due to reduction or loss of biological productivity” to help clarify what is meant by degradation.

Please provide assumptions that motivate the experimental manipulation, e.g., how this diversity loss is realistic reflecting a scenario of soil degradation. Best to do this early on, for example at the end of the Introduction.

We have now moved the assumptions for the experimental manipulation that was in the methods up to the last paragraph of the Introduction. “The exclusion of organisms based on size can lead to a functional simplification of soil communities as body size is directly associated with trophic guilds, metabolic rates, population density and generational turnover (Bradford et al., 2002, Yodzis and Innes 1992, Woodward et al., 2005, Coudrain et al., 2016). Furthermore, the size-based reduction of soil organisms parallels the impact of land management practices, such as soil tillage, that physically damage soil organisms depending on their size and thus, also disrupting the community structure of soil biota (Jansa et al., 2003, Postma-Blaauw et al., 2010, Köhl et al., 2014, Wagg et al., 2018).”

Define stability and state how it was calculated.

We have now defined temporal stability: “we hypothesize that greater soil biodiversity should also maintain a greater and less variable ecosystem functioning through time and thus stabilize multiple ecosystem functions.” and: “This enabled us to assess effects of the soil microbial community on the temporal stability of ecosystem functioning, which we calculated as invariability, the inverse of the coefficient of variation of ecosystem functions across all five time points (Tilman et al., 2006; Pimm, 1984; Tillman, 1999).”

Define temporal asynchrony, and "asynchronous dynamics". Because of such dynamics, microbes are less redundant. We do agree that this is important, but the argument here sounds more like interpretation.

Asynchrony is now clarified at the end of the first paragraph of the Introduction: “Thereby the asynchronous temporal fluctuations among species in their abundance and contribution to various ecosystem functions in more diverse communities contributes to a greater overall ecosystem functioning over time (Hautier et al., 2018; Isbell et al., 2011, Yachi and Loreau, 1999).”

We have also reworded the sentences: “However, functional redundancy is likely to fade as multiple time points and ecosystem functions are considered (Hector and Bagchi, 2007; Isbell et al., 2011, Hautier et al., 2018). Considering that soil biodiversity has been shown to support numerous ecosystem functions, we hypothesize that greater soil biodiversity should also maintain a greater and less variable ecosystem functioning through time and thus stabilize multiple ecosystem functions. Specifically, microbial taxa may fluctuate asynchronously through time where different taxa support different functions at different times, thus providing insurance that some of these microbial taxa will be present at any given time to stabilize the functioning of the ecosystem (Isbell et al., 2011; Yachi and Loreau 1999; Mori et al., 2016)”